# Therapeutic Potential of Honey and Propolis on Ocular Disease

**DOI:** 10.3390/ph15111419

**Published:** 2022-11-17

**Authors:** Norhashima Abd Rashid, Siti Nur Farhana Mohammed, Syarifah Aisyah Syed Abd Halim, Norzana Abd Ghafar, Nahdia Afiifah Abdul Jalil

**Affiliations:** 1Department of Biomedical Science, Faculty of Applied Science, Lincoln University College, Petaling Jaya 47301, Malaysia; 2Department of Anatomy, Faculty of Medicine, Universiti Kebangsaan Malaysia, Kuala Lumpur 56000, Malaysia

**Keywords:** honey, propolis, natural product, pharmacological, antioxidants, anti-inflammatory, antimicrobial, analgesic, ocular disease

## Abstract

Honey and propolis have recently become the key target of attention for treating certain diseases and promoting overall health and well-being. A high content of flavonoids and phenolic acids found in both honey and propolis contributes to the antioxidant properties to scavenge free radicals. Honey and propolis also exhibited antibacterial effects where they act in two ways, namely the production of hydrogen peroxide (H_2_O_2_) and gluconic acids following the enzymatic activities of glucose oxidase, which exerts oxidative damage on the bacteria. Additionally, the anti-inflammatory effects of honey and propolis are mainly by reducing proinflammatory factors such as interleukins and tumor necrosis factor alpha (TNF-α). Their effects on pain were discovered through modulation at a peripheral nociceptive neuron or binding to an opioid receptor in the higher center. The aforementioned properties of honey have been reported to possess potential therapeutic topical application on the exterior parts of the eyes, particularly in treating conjunctivitis, keratitis, blepharitis, and corneal injury. In contrast, most of the medicinal values of propolis are beneficial in the internal ocular area, such as the retina, optic nerve, and uvea. This review aims to update the current discoveries of honey and propolis in treating various ocular diseases, including their antioxidant, anti-inflammatory, antibacterial, and anti-nociceptive properties. In conclusion, research has shown that propolis and honey have considerable therapeutic promise for treating various eye illnesses, although the present study designs are primarily animal and in vitro studies. Therefore, there is an urgent need to translate this finding into a clinical setting.

## 1. Introduction

Honey is naturally sweetened food that is collected by honeybees from floral nectar and sometimes from insect secretions called aphids [1]. Most of the honey content is made up of sugars, predominantly fructose and glucose, with modest amounts of fructo-oligosaccharides [1]. Most of the pharmacological properties of honey, including antibacterial, analgesic, anti-inflammatory, antioxidant, and immunomodulatory, are attributable to the presence of flavonoids (pinocembrin, kaempferol, quercetin, galangin, apigenin, and chrysin) and phenolic acids (ellagic, p-coumaric, ferulic, gallic, benzoic, and rosmarinic acids) [2]. Most of these compounds work synergistically to provide a wide spectrum of biological capabilities [2]. Figure 1 shows the flavonoids compound in honey.

On the other hand, propolis is used by honeybees as a medium to patch any cracks or fragments found in the hive to help prevent the predator’s attack. Propolis functions as a thermal insulator to maintain the interior temperature of beehives at 35 °C. In addition, propolis hardens the cell wall, contributing to the aseptic conditions in the hives [3]. Propolis is a balsamic and resinous compound secreted by bees that are made up of a mixture of 50% plant resins, 30% waxes, 10% essential and aromatic oils, 5% pollen, and 5% other organic components [4]. Among the flavonoids found in propolis are pinocembrin, acacetin, chrysin, rutin, luteolin, kaempferol, apigenin, myricetin, catechin, naringenin, galangin, and quercetin; two phenolic acids, caffeic acid and cinnamic acid; and one stilbene derivative called resveratrol has been found in propolis extracts [5]. Propolis is also enriched with essential vitamins such as B1, B2, B6, C, and E and beneficial minerals such as magnesium, calcium, potassium, sodium, copper, zinc, manganese, and iron [6]. Phenolic compounds that contribute to a wide range of health-promoting benefits can be found practically in all propolis forms, irrespective of their geographical origin and season [5]. Figure 1 and Figure 2 show some flavonoids and phenolic acids found in honey and propolis.

Recently, researchers across the globe have discovered the medicinal value of many other natural products [7,8,9,10], including honey and propolis, in various eye diseases. Honey is commonly used to treat external ocular disease models affecting the cornea, conjunctiva, and eyelids, particularly in the form of topical application. Any defects in these structures may compromise the ocular integrity and lead to possible visual-threatening conditions, such as microbial infections, chronic inflammation, and ulceration, that may potentially entail eye blindness if not treated sufficiently. Hence, prompt and appropriate treatment must be imposed in managing eye diseases [11,12]. To date, eye diseases are conventionally treated with topical eye drops or eye ointment. However, the preparation of topical eye drops frequently use a preservative, namely benzalkonium chloride, that has consistently been linked to some alarming adverse effects such as dryness of the eye, damage in the epithelial barrier, disruption of the ocular fluid drainage, and eye inflammation, especially in cases with long term exposure [13,14,15]. This has increased the propensity of studies to find nature-based alternatives, such as honey, to treat eye diseases.

Propolis, contrarily, has more favorable effects in the interior area of the eye, such as retinal ganglion cells, optic nerve, and uvea [16]. The medicinal properties of propolis have been acknowledged since ancient times [6]. Nano-preparation of propolis-acetazolamide exerted hypotensive effects on glaucoma. At the same time, the high content of hydrocarbon in propolis conferred neuroprotection on the protein of the optic nerve secondary structure [17], giving hope for the prevention of blindness in glaucoma. Recently, propolis has been shown to provide a neuroprotective effect on retinal ganglion cells through the downregulation of apoptosis and inflammatory pathways [18]. The purpose of this review is to establish the advantages of antibacterial, anti-inflammatory, antioxidant, and analgesic properties of both honey and propolis. In addition, based on recent studies, we included a list of possible eye disorders that can be treated with honey and propolis.

## 2. Medicinal Properties of Honey

### 2.1. Antioxidant

The presence of phenolic acids, flavonoids, ascorbic acids, proteins, and carotenoids in honey contributes to its antioxidant properties [19]. Antioxidants prevent damage from oxidants such as O_2_, OH-, superoxide, and/or lipid peroxyl radicals. The imbalance between free radicals and antioxidant agents causes oxidative stress [19]. Inflammation, infection, and cancer are susceptible to oxidative stress. The defense system of cells produces free radicals and oxidative protective substances, including superoxide dismutase, peroxidase, catalase, ascorbic acid, and polyphenols [20]. 

Many studies on honey from diverse floral origins and regions have demonstrated the great antioxidant characteristics of honey [21]. Phenolic acids of honey protect against DNA damage by chelating ferrous ions and scavenging hydrogen peroxide [22]. Additionally, phenolic acids scavenge reactive oxygen and nitrogen species, in addition to the deactivation of peroxyl radical, hypochlorous acid, and nitric oxide [21]. Melanoidins, products of the Maillard reaction, were found to be the main constituents for the radical scavenging capacity of honey [23]. The antioxidant activity of gallic acid involves the Nrf2-antioxidant response element signaling pathway [24]. Gallic acid also suppressed oxidative stress by modulating Nrf2-HO-1-NF-κB signaling pathways [25].

Chrysin reduced ROS, malondialdehyde levels, and lactate dehydrogenase release and improved catalase activity as part of antioxidant mechanisms [26]. Chrysin upregulated HO-1, GCLC, and GCLM gene transcription by modulating ERK2/Nrf2/ARE signaling pathways to inhibit oxidative stress [27]. Apigenin counteracts oxidative stress by modulating redox signaling pathways (Nrf2, MAPK, and P13/Akt) [28]. Table 1 summarizes the antioxidant signaling pathways of phenolic acids and flavonoids. 

### 2.2. Antibacterial

The antibacterial properties of honey have been widely used for treating and preventing wound infections for many years [30]. Primarily, the antibacterial effects of honey have been associated with two theories: peroxide and non-peroxide activities [31]. In peroxide theory, the production of hydrogen peroxide (H_2_O_2_) and gluconic acids following the enzymatic activities of glucose oxidase in honey exerts oxidative damage on the bacteria. This happens when the H_2_O_2_ degrades the bacterial deoxyribonucleic acid (DNA), inhibiting its growth [23,32]. The presence of gluconic acid in honey together with H_2_O_2_ as a result of oxidation of oxygen upon dilution has been found to be part of the antimicrobial property of honey. Both components exhibit synergistic antibacterial mechanisms by affecting the polarity of the cell membrane and the integrity of the cell wall [33].

The non-peroxide theory describes the contribution of various essential flavonoids and phenolic compounds that exert their individual bacterial-fighting mechanism. These are in addition to the physicochemical properties and inert antibiotic properties of honey, including the low acidic pH, high sugar content, and presence of antimicrobial peptides such as bee defensin-1 and methylglyoxal (MGO) phytochemical components in honey [34,35]. Some of their antimicrobial actions are summarized in Table 2, Table 3 and Table 4. For instance, ferulic aid disrupted bacterial membranes, causing structural and functional alteration [36]. Additionally, the MGO and its precursor, dihydroxyacetone (DHA), hindered bacterial growth by inhibiting urease, an important enzyme in bacteria for acclimatization and survival in acidic conditions with the production of ammonia [37]. 

A novel *Hovenia dulcis* monofloral honey demonstrated great antibacterial activity against common foodborne microbes, namely *Listeria monocytogenes*, *Staphylococcus aureus*, *Salmonella Typhimurium,* and *Escherichia coli* [53]. Honey–chitosan hydrogel formulation used in burn wounds has been shown to inhibit *Pseudomonas aeruginosa*, *Staphylococcus aureus*, *Klebsiella pneumonia*, and *Streptococcus pyogenes* growth [54]. Similarly, during the perioperative phase, a topical honey-based eye drop prepared from monofloral honeydew honey inhibited the growth of both Gram-positive and -negative bacteria [55].

Surprisingly, the honey-resistant microbial strain has never been reported to date. In fact, honey has illustrated its broad-spectrum antibacterial capacity towards various aerobes, anaerobes, Gram-positive and -negative bacteria, even against the well-known multi-drug resistant bacteria, such as *S. aureus*, *P. aeruginosa,* and *Enterococci*. Therefore, using honey as an alternative in conditions with a rising emergence of antimicrobial resistance, such as burns and thermal wound infections by methicillin-resistant *S. aureus* (MRSA), *Enterococcus* sp., and vancomycin-resistant *Enterococci* has been highly advocated [56]. These positive effects were thought to be due to the heterogeneous therapeutic properties of honey, including antibacterial and anti-inflammatory effects [56,57]. 

Recent case series reported by Nair et al. (2020) has illustrated the enormous impact of medical grade honey (MGH) substitution over the standard treatments for infected diabetic foot ulcers, such as antibiotics, silver dressing, and maggot therapy. MGH therapy has generally eliminated multi-resistant *P. aeruginosa* and *Streptococcal* bacterial infections, hastening the wound healing process and averting the risk of amputation [58]. This remarkable efficacy of honey in inhibiting MRSA and Methicillin-sensitive *S. aureus* infections was similarly observed in a previous study [57] and other laboratory-based studies [59,60]. Moreover, honey effectively impeded the biofilm matrix creation by *P. aeruginosa*, which was considered the key factor in establishing the pathogen sustainability and resistance of antibiotic agents in long-standing incurable wounds [61]. The synergism of medically approved Manuka honey in combination with oxacillin exhibited hindrance of MRSA growth and restored sensitivity towards oxacillin. These effects were attributed to the downregulation of the mecR1 gene product, a transducer for antibiotic resistance in MRSA, seen in cells treated with Manuka honey [62].

### 2.3. Anti-Inflammatory

Inflammatory pathways impact the pathogenesis of a number of chronic diseases, such as type 2 diabetes mellitus (T2DM) [63,64], which involves common inflammatory mediators and regulatory pathways. Inflammatory stimuli activate intracellular signaling pathways that trigger inflammatory mediators’ production. 

Various studies have evinced the promising anti-inflammatory benefits of honey that could nominate it as a prospective non-pharmaceutical alternative in managing inflammatory conditions [34,65]. During inflammation, there is an augmentation in the numerous proinflammatory factors comprising cytokines such as interleukins (IL-1, IL-6 and IL-10; tumor necrosis factor alpha (TNF-α)); and inflammatory enzymes such as cyclooxygenase (COX), lipoxygenase (LOX) and many others. Additionally, there is also infiltration of various inflammatory cells, including monocytes, macrophages, and leukocytes [19,34]. It has been inferred that these out-turns have resulted from the activation of inflammatory pathway components, namely the mitogen-activated protein kinase (MAPK) and nuclear factor kappa B (NF-kB) that regulate the downstream inflammatory mediators [19]. Generally, any alteration in the physiology of inflammation produced by honey may suggest their anti-inflammatory effects.

Similar to the antibacterial effects, flavonoids and phenolic acids in honey are the major contributors to its anti-inflammatory properties [66,67]. Galangin suppressed the activation of NF-κB and MAPK signaling hence reducing the associated inflammatory mediators’ secretion, including nitric oxide (NO), inducible NO synthase (iNOS), and IL-6 [68]. At the same time, chrysin attenuated the proinflammatory activity of IL-1β and TNF-α [69]. Quercetin inhibited the arachidonic acid cascade by hindering the inflammatory COX and LOX enzymes, which subsequently decreased the inflammatory mediators’ end products such as prostaglandins and leukotrienes [70].

A combination of three natural kinds of honey (Trihoney) lessened the serum inflammatory cytokines of TNF-α, IL-1β, and IL-6 in a hypercholesterolemic animal model [71]. A favorable limitation on NO production has been observed following the Kelulut honey use [72], while Manuka honey [73] and Safflower honey [74] were both noted to impede the production of TNF-α, IL-1β, IL-6, and iNOS as well as suppress the NF-kB pathway in an in vitro lipopolysaccharides-stimulated inflammation [73]. Likewise, similar results were observed in Manuka honey treatment on the acetic acid-induced gastric ulcer rats that suppressed TNF-α, IL-1β, and IL-6 secretions along with augmentation of anti-inflammatory cytokine IL-10 level [75]. Additionally, honey decreased the release of inflammatory cells such as macrophages, monocytes, and leukocytes, subsequently limiting their activities on producing reactive oxygen species and inflammatory mediators, hence minimizing inflammation [76]. A series of in vivo studies demonstrated that honey treatment in carrageenan-induced inflammation rats showed a significant reduction in the size of the paw edema and the proinflammatory factors (IL-6, TNF-α, NO, and prostaglandin E2 (PGE2). Honey also suppressed gene expression of NF-kB in paw tissues, thus decreasing the subsequent inflammatory mediators of COX-2 and TNF-α [77,78]. 

### 2.4. Anti-Nociceptive 

Honey has the potential as an analgesic because it eases pain via modulation at a peripheral nociceptive neuron or binding to an opioid receptor in the higher center. Pain from the periphery is transmitted through Aδ or C fibers synapse initially at the dorsal horn of grey matter, the primary afferent neuron. Subsequently, endogenous mediators, including substance P, bradykinin, serotonin, histamine, and prostaglandin, are released to stimulate peripheral nociceptive neurons. Later, the pain is transmitted to the higher center, the somatosensory cortex, via the spinothalamic tract [79]. Different types of honey have exhibited various ways of pain-relief mechanisms, as summarized in Table 5.

Yemeni Sidr honey showed anti-nociceptive effects in a few experimental pain models: acetic acid and formalin-induced writhing, histamine, and carrageenan-induced paw edema in an experimental rat model. Acetic acid generated pain indirectly by releasing mediators that stimulated peripheral nociceptive neurons to increase vascular permeability, reduce nociceptor threshold and stimulate the nervous terminal of nociceptive fibers. Pre-treatment with Yemeni Sidr honey displayed a reduction in acetic acid-writhing response, as Yemeni Sidr honey reduced the release of inflammatory mediators and consequently blocked the peripheral nociceptive effect. Moreover, formalin produced pain via two distinct phases: the first phase was the transient neurogenic pain phase, which was inflicted by a direct effect on the C nerve fiber; the second phase was inflammatory pain, as formalin activated inflammatory response and released nociceptive mediators. After the Yemeni Sidr honey administration, a significant reduction in formalin-induced flinching and licking during the inflammatory phase was observed. In the histamine and carrageenan-induced paw edema models, Yemeni Sidr honey significantly reduced the paw edema volume as it inhibited the release of inflammatory mediators, including histamine, serotonin bradykinin, and prostaglandin [80].

Mad honey is a nectar–pollen mixture of honey and belongs to the Rhododendron species [87]. Grayonotoxin found in mad honey has been linked to the analgesic benefit. Administration of mad honey in the normoglycemic mice elevated the thermal pain threshold latency. In diabetic mice, mad honey reduced and restored the thermal pain threshold to normal. The ability of mad honey to suppress diabetic neuropathic pain was via the binding of grayanotoxin to the sodium channel, leading to modification of that sodium channel gating and release of gamma-amino-n-butyric acid (GABA) from isolated nerve terminal [81,82]. 

Tualang honey is a multi-floral jungle honey produced by *Apis dorsata* [88]. The potentiation of Tualang honey as an analgesic agent might be attributed to its anti-inflammatory and antioxidant properties and partly contributed by its action on opioid receptors [83]. Additionally, topical application of Tualang honey in post-tonsillectomy wounds has been shown to reduce the mean pain score and frequency of awakening during the night, possibly due to the soothing effect of topical application of honey on the tonsillar bed mucosa [84]. 

Nigerian honey has demonstrated its analgesic effects via the action on the opioid receptors, whereby the pain-relieving mechanism via its action on the opioid receptors, whereby its pain-relieving activity was abolished upon administering Naloxone, an opioid receptor antagonist [85]. Honey also attenuated the pain progress in chronic illnesses such as osteoarthritis (OA). Oral administration of honey significantly increased the paw withdrawal threshold in OA rats [86]. Furthermore, honey reduced inflammatory markers and vascular endothelial growth factor (VEGF), which were increased during osteoarthritic disease progression. The anti-nociceptive effect of honey was closely related to its anti-inflammatory and antioxidant capacity, as it was vital for recovery and preventing the progression of the disease [86]. Gelam honey, a farm bee honey, inhibited NO and PGE2, and reduced histamine and cytokine release in the inflamed paw edema, subsequently inducing a pain-relief effect [89]. 

## 3. Medicinal Properties of Propolis

### 3.1. Antioxidants

The antioxidant properties of propolis are due to the presence of bioactive compounds such as flavonoids and phenolic acids. The estimation of its antioxidant activity was confirmed using DPPH, ABTS+, FRAP, and ORAC assays. According to published data, propolis extracts typically have total phenolic contents of 30 to 200 mg of gallic acid equivalents (GAE)/g of dry weight, flavonoid contents of 30 to 70 mg of quercetin equivalents (QE)/g, and DPPH-free radical-scavenging activities of 20 to 190 g/mL [88]. Brazilian green propolis’ strong antioxidant activity is primarily related to its high phenolic content [90]. Contrary to Brazilian propolis, the antioxidant activity of poplar propolis seems to be significantly impacted by both its total polyphenol and total flavonoid levels [91]. The amount and content of the bioactive compound of the propolis vary as it is largely influenced by many factors such as bee species, season, temperature, and geographical location [92]. Therefore the antioxidant activity, free radical scavenging activity, and the ability to inhibit lipid peroxidation of the propolis are also dependent on the aforementioned factors.

The effect of the consumption of commercially available propolis solution (Beepolis^®^) on the oxidative state and lipid profile in a human population in Chile was examined by Mujica et al. [93]. These antioxidant effects of propolis give tremendous health benefits to humans. Following administration of 15 drops of Beepolis^®^ twice daily for three months has shown a 67% reduction in thiobarbituric acid reactive substances (TBARS) and an increase in the level of reduced glutathione (GSH) and HDL concentration in the studied population. Another study showed a remarkable increase in SOD activity in healthy male participants who consumed daily commercially available powdered propolis extract for 30 days [88]. 

The effect of Brazilian green propolis supplementation on the antioxidant levels of type 2 diabetes Mellitus (T2DM) patients was investigated. Oral administration of 900mg daily for 18 days revealed increased serum levels of GSH, total polyphenols, and IL-1 and IL-6 with a significant reduction in TNF-serum levels. However, the improvement of antioxidant parameters did not reflect on the diabetes markers such as blood glucose, glycosylated hemoglobin, and insulin level [94].

### 3.2. Antibacterial

Propolis tends to interfere with the bacteria’s pathogenic potential by reducing adenosine triphosphate (ATP) generation, impairing bacterial cell membrane permeability, disrupting membrane potential, retarding bacterial motility, and hindering bacterial ribonucleic acid (RNA) and DNA production [4]. The susceptibility of various bacteria to different kinds of propolis is shown in Table 5.

Gram-positive bacteria are found to be substantially destroyed by propolis, whereas Gram-negative bacteria are more likely to demonstrate resilience [4,95,96]. This was possibly due to the barrier created by the negatively charged lipopolysaccharide on the Gram-negative bacterial wall. Additionally, the presence of hydrolytic enzymes weakens the active compounds in the propolis [4,95,96]. Nonetheless, there is an exception where Nepalese propolis showed similar antibacterial activity against both Gram-negative and positive bacteria [97]. 

Flavonoids and polyphenols are the main contributing factors for most antibacterial activities in propolis (Table 6). Biochemical analysis of propolis extracts from *Apis mellifera* L. and *Trigona* sp. contained pterocarpans and flavonoid aglycones (mostly neoflavonoids and isoflavonoids). By using the disc diffusion test, both propolis showed the highest antibacterial activity against *H. pylori*, *S. aureus*, and *S. flexneri* [97]. Pinocembrin and apigenin in Chilean propolis exerted powerful antibacterial and antibiofilm even at low concentrations. The effect of polyphenols in Chilean propolis seems attributable to combine mechanisms, not solely limited to antimicrobial potential, as a considerable reduction in the biofilm thickness and cellular adhesion was also observed [98].

Propolis is believed to inhibit protein synthesis and cause partial bacterial lysis. Together with antibiotics, propolis might enhance the antibacterial effect and shorten the healing period [95]. On the other hand, synergism between propolis and several antibiotics was verified in many studies [97,111]. A combination of Chilean propolis with antibiotics such as amikacin and tetracycline exhibited synergism, especially against *S. aureus* [97]. In addition, the synergy between Brazilian and Bulgarian propolis and antibiotics (chloramphenicol, tetracycline, and neomycin) resulted in bacteriostatic and bactericidal effects, respectively, which acted on the ribosome against *S. Typhi* [111].

### 3.3. Anti-Inflammatory

Propolis, such as honey, has anti-inflammatory properties due to its numerous bioactive components. One of the polyphenols contained in propolis is caffeic acid. In lipopolysaccharides-induced inflammation, both propolis and caffeic acid suppressed NO production in macrophages by downregulating NF-κB pathways and attenuating p38 MAPK and c-Jun N-terminal kinases 1/2 phosphorylation [112]. Ethyl ester of arachic acid extract of propolis originated from Tala-Mokolo, Cameroon, possessed anti-inflammatory effects in both acute and chronic phases. Arachic acid ethyl ester profoundly inhibited the edema production in the paws and ears of the rat model, suggesting the potential role of this compound to suppress the synthesis and release of inflammatory mediators, including bradykinin, serotonin, prostaglandins, and histamine or by inhibiting cyclooxygenase pathway [113]. Cinnamic acid derivatives present in the Brazilian propolis extract, such as caffeic acid phenethyl ester (CAPE) and artepillin C, inhibited IL-17 synthesis in cultured murine splenocytes by lowering retinoic acid-related orphan receptor gT expression. Other compounds of Brazilian propolis, including baccharin, culifolin, and drupanin, suppressed inflammatory signaling in murine RAW 264.7 macrophages, thus lowering TNF- and/or IL-6 production. Together, the anti-inflammatory activity of Brazilian propolis is mediated in part through the control of Th17 differentiation and macrophage activation by cinnamic acid derivatives [114].

### 3.4. Anti-Nociceptive 

The analgesic effect of propolis has been studied extensively in several animal models. In a tail flick experiment, a water extract of Anatolian green propolis produced a strong analgesic effect. Supplementation of this propolis to the toothpaste formulations has been shown to have analgesic activity and could be used as a component in treating periodontal disease [115].

Phenolic acids such as caffeic acid and high level of flavonoids such as galangin, pinocembrin, and chrysin found in Chinese propolis has been linked to the anti-nociceptive effect of propolis. Distinct fractions of Chinese propolis extract richer in polyphenolic constitutions exhibited centrally and peripherally anti-nociceptive effects, which could be associated with their antioxidant activities. These findings support the clinical use of propolis as a treatment option for painful diseases [116].

The hydroalcoholic extract of red Brazilian propolis (HERP) attenuated the abdominal constrictions induced by acetic acid at a lower dose in comparison to formononetin, a biomarker of HERP. This could be related to the other HERP elements that have anti-nociceptive/anti-inflammatory actions in the peripheral nervous system [117]. The release of inflammatory mediators such as serotonin, prostaglandins, bradykinins, and cytokines (TNF-α, IL-1β, and IL-8) was due to the activation of acid-sensitive ion channels and the transient receptor potential vanilloid one cation channels [118].

Moreover, HERP suppressed both early and late phases of inflammatory pain in formalin-induced nociception, whereas formononetin showed better inhibition in the early phase. Formononetin reduced glutamate-induced nociception in the same way that 30 mg/kg of HERP did. The open field test showed no significant alterations following HERP treatment, while formononetin attenuated spontaneous motor behavior. Furthermore, the anti-nociceptive effects of HERP on inflammatory and neurogenic pain caused no motor side effects, which could be attributed to the other compounds in the extract [117].

## 4. Medicinal Values of Honey-Related Products on Ocular Diseases

By considering the existing treatment modalities for eye diseases, the administration of honey is mainly targeted for its topical use. Therefore, studies involving honey in eye diseases have been focusing more on eye drops and ointment [57]. Among the eye diseases that have been incorporated with the beneficial honey application are typically conditions affecting structures of the ocular surface, mainly the conjunctiva, cornea, and eyelid.

At present, different types of honey have been tested clinically for use in the medical field, but they are yet to achieve medical-grade status. Currently, the available MGH for use in ophthalmology is a Manuka-based (Optimel) eye drop that was indicated for the treatment of chronic dry eye and blepharitis [57]. 

### 4.1. Conjunctivitis 

Conjunctivitis is defined as an inflammation of the conjunctiva that can either be due to infectious (microbial pathogens) or non-infectious, such as allergy causes. It is one of the commonest causes of non-emergent eye-related visits to the emergency department, accounting for 28% of the cases [119]. A remedy that can address the wide range of aetiologies together with the concomitant inflammation is advantageous such as eye drops with a combination of corticosteroid and broad-spectrum antibiotics [120]. However, the known harmful ocular risks associated with the use of corticosteroids, such as increased intraocular pressure, cataract, and corneal ulcers, has given rise to the search for an alternative natural agent that renders similar or even better therapeutic effects [121]. 

Topical application of stingless bee honey in *S. aureus* and *P. aeruginosa* induced conjunctivitis for 12 hours for two weeks reduced the inflammatory signs, duration of infection, and time for complete resolution of bacterial infection. This finding was comparable with the gentamicin-treated group [121]. In a double-blind clinical trial in vernal keratoconjunctivitis, adjuvant therapy of 60% honey-based topical eye drop has resulted in a reduction in eye redness and limbal papillae, with promising minimal use of steroids and minimal increase in eye pressure [122]. Given that vernal keratoconjunctivitis is an allergic inflammatory eye disease, honey may have helped to alleviate the symptoms by reducing inflammatory reactions [122].

### 4.2. Keratitis

Keratitis is an inflammation of the cornea, which may or may not be associated with an infection. Defects in the protective layer of corneal epithelium interfere with its defense mechanism and render infiltration of the pathogenic microbes, resulting in corneal inflammation [123]. This potentially visual-threatening infectious keratitis can either be caused by (a) bacterial organisms, commonly the coagulase-negative *Staphylococcus* and *P. aeruginosa* [124]; (b) virus, predominantly the Herpes Simplex Virus; or (c) fungal infection frequently by Candida species [125,126]. Microbial keratitis is among the leading cause of blindness across the globe [127]. 

At present, infectious keratitis is conventionally treated with the respective antimicrobial medication based on the underlying causative agents, with an additional topical steroid application in cases of interstitial keratitis [124,128]. Interstitial keratitis is described as the inflammatory reaction of corneal stroma that occurs following the host’s reaction to bacterial, viral, or parasitic antigens elsewhere or as a result of an autoimmune response without an apparent corneal infection [128]. 

Microbial keratitis caused by *P. aeruginosa* can manifest as a suppurative infiltration that can worsen to form corneal perforation and melting, causing blindness. This may require long-term care and expensive treatment. Contact lens wearers are particularly vulnerable to *P. aeruginosa* keratitis. Ring abscess is a distinguished feature of *P. aeruginosa* keratitis due to the ring-shaped aggregation of polymorphonuclear leukocytes forming a ring with a central corneal lesion. Occasionally, serrated or satellite lesions are also present together with ring abscesses [129]. The pathogenicity of *P. aeruginosa* is due to its large genome [130] and the presence of various virulence factors [130], which are mediated by a complex regulatory network and synergistically infect the host tissue [131,132]. Moreover, during infection, inflammatory cytokines such as IL-1, IL-6, and IL-8 via NF-κB [133] are overproduced, which potentially causes corneal damage, scarring, and even blindness [134]. Additionally, increasing antibiotic resistance to *P. aeruginosa* is a major health concern [135] that urges research into alternative therapy. 

In a rabbit model of Pseudomonas-induced keratitis, topical Tualang honey 30% displayed similar results to topical gentamicin and a combination of both in terms of clinical and antimicrobial effects. Clinically, all three treatments (Tualang honey, gentamicin, and mixed of both) improved conjunctival hyperemia though no apparent effects on corneal edema. Slit lamp examination score, a collective score of corneal infiltrates, corneal ulcer, hypopyon, and corneal perforation, were almost similar. Additionally, the Tualang honey and gentamicin mixture revealed the lowest bacterial growth, yet no marked difference between the single treatment of each [136].

Another study reported honey’s anti-inflammatory and antiangiogenic effects on the Pseudomonas endotoxin-induced keratitis model. Topical honey application from soybean and wildflower origin inhibited cytokines, namely TNF-⍺, IL-12, and angiogenic factor TGF-β. These findings supported the histological results, where fewer inflammatory cells, such as neutrophils, were found in the stromal limbus of the honey-treated group [137]. A separate study using 90% natural honey was found to be as effective as topical 0.3% ciprofloxacin in stromal keratitis infected with *P. aeruginosa*. The natural honey prevented the progression of the size or depth of the corneal ulcer, conjunctival inflammation, and pus discharge in addition to suppression of the bacterial growth, comparable to those treated with ciprofloxacin [138].

### 4.3. Blepharitis

Blepharitis is a chronic inflammatory condition of the eyelids that can be classified into anterior and posterior blepharitis. Anterior blepharitis involves the anterior lid margin and eyelashes, while posterior blepharitis occurs following the meibomian gland dysfunction (MGD) [139,140]. Its underlying mechanism is yet to be fully understood, but it is generally associated with the microbial colonization of common ocular bacteria, such as *S. aureus*, or parasitic infestation with Demodex mites that lead to subsequent inflammatory cascades [141]. With the limited knowledge of its pathophysiology, management of this condition has been focused on reducing the associated infection and inflammation with a substantial improvement in signs and symptoms of blepharitis after the applications of antibiotics and topical corticosteroids [142,143]. However, concerns pertaining to the development of antibiotic resistance and anti-inflammatory medications’ adverse effects in long-term usage have prompted the need to find an alternative treatment [140]. 

A preclinical study on the cyclodextrin-complexed MGO Manuka honey microemulsion (MHME) has shown to be effective in inhibiting bacterial growth in blepharitis. The in vitro study found that *S. aureus* and *S. epidermidis* growth were suppressed at doses of 400 mg/kg and 550 mg/kg of MGO. Subsequent instillation of either diluted MGO MHME or saline control in rabbit eyes revealed the safety and tolerability of the MHME. During this in vivo phase, no significant immediate or cumulative harmful effects were identified upon the evaluation of tear film osmolarity, lipid layer grade, tear evaporation rate, fluorescein staining, phenol red thread, corneal opacity, conjunctival hyperemia and iris appearance grades [144]. Therefore, MHME was further tested on human subjects in the form of eye cream for periocular application for a two-weeks duration. In this study, the MHME eye cream caused no significant changes in clinical (visual acuity, eyelid irritation, ocular surface characteristics) and impression cytology evaluation (matrix metalloproteinase-9, IL-6, and MUC5AC) [145], implying the safety and tolerability of MHME eye cream in the clinical study despite transient ocular stinging, which disappeared after water irrigation. 

As MHME eye cream is safe and well tolerated, a subsequent randomized, masked clinical study was initiated for a period of 3 months. Topical overnight application of MHME cream significantly reduced SANDE and SPEED symptomology scores (dry eye questionnaire) in treated eyes on days 30 and 90. Clinical improvement in lipid layer thickness, tear film stability, and inferior lid wiper epitheliopathy was reported on day 90. During a 3-month therapy period, Ocular Demodex, Propionibacterium acne, Corynebacterium macginleyi, and *S. epidermidis* growth dropped considerably. The therapeutic benefits of the topical MHME eye cream for blepharitis are likely due to multifactorial factors, particularly the anti-demodectic, antibacterial, and anti-inflammatory activities of cyclodextrin-complexed MGO MHME [145].

### 4.4. Corneal Injury

Corneal ulcers and abrasions are categorized as eye emergency conditions [119,146] that require an immediate, holistic approach to prevent the possible complications that may result in unwanted ocular morbidities. The main underlying pathological mechanism is a breach or defect in the corneal structure, which either involves only the superficial epithelial layer in corneal abrasion [11] or may extend up to the stromal layer in corneal ulcer [147]. Corneal injuries often result from mechanical trauma [12], such as from the use of contact lenses or the presence of foreign bodies or fingernail scratches, and can also be attributable to an infective etiology [147,148]. The favorable effects of honey on skin wound healing are very well documented. This has encouraged researchers to embark on its promising effects on the cornea. Although these studies are mainly conducted using in vitro corneal cells, there is still a scarcity in the literature on the effects of honey on in vivo models and clinical studies.

Proliferation is one of the primary steps in wound healing to repopulate the injured area [148]. In a study by Ker-woon et al. (2014), Acacia honey (AH), which is a monofloral honey yielded by *Apis mellifera* honeybees, has been studied for its proliferative capacity on corneal epithelial cells (CEC). AH enhanced the proliferation of CEC while preserving its normal histological features and cell cycle, accompanied by increased DNA content and cell nuclei [149]. In subsequent in vitro corneal abrasion models, AH accelerated the wound closure, likely attributed to the additional ATP supply of AH. Moreover, AH upregulated both genes and proteins expression of cytokeratin-3, a cluster of differentiation 44 (CD44), and fibronectin [150]. Fibronectin and CD44 expressions were enhanced during the initial and middle phases of the experiment, respectively. This is in accordance with the function of fibronectin in the early phase of wound healing as a temporary extracellular matrix to aid cellular migration [151]. Similarly, CD44 aids the migratory phase by providing adhesive durability for cell-to-cell and cell-to-matrix interactions [150].

Poorly treated corneal abrasion can progress into a corneal ulcer affecting the function of the dormant keratocytes in the corneal stromal layer [152], or otherwise, the initial injury may also penetrate directly deep into the stroma. Keratocytes are stromal cells derived from mesenchymal cells that are essential for corneal transparency, maintenance of its shape, and production of the extracellular matrix [153]. The healing process of the corneal stromal layer activates quiescent keratocytes and triggers their transition into repair phenotypes: fibroblasts and myofibroblasts [154]. These different phenotypes have specific gene and protein expressions such as aldehyde dehydrogenase (ALDH), α-smooth muscle actin (α-SMA), vimentin, collagen I, and lumican, which can be analyzed for their presence and functions. The transition of quiescent keratocytes to fibroblasts and myofibroblasts caused reduced ALDH expression, one of the essential corneal crystallins members involved in corneal transparency [155]. The diminished presence of α-SMA is associated with corneal fibroblast differentiation to myofibroblasts during wound closure [156], while increased expression of vimentin indicated proliferation during wound healing [156]. Moreover, lumican is believed to retain corneal transparency during stromal wound healing through its role as a proteoglycan in collagenous matrix assembly [156,157]. In an in vitro corneal ulcer wound healing model, 0.0025% concentration of AH has been reported to expedite wound closure. AH upregulated vimentin, collagen I, and lumican gene expressions while downregulated both gene and protein expressions of ALDH, ⍺-SMA, and matrix metallopeptidase 12. Hence, AH expedited stromal wound closure by enhancing cellular migration and differentiation [156]. 

Along with AH, Gelam honey (GH) is another type of Malaysian honey that is collected by the *Apis mellifera* honeybees from Gelam (*Melaleuca* spp) trees [158]. Apart from its high sugar content, GH contains a substantial amount of vitamins C, B1, and B3; flavonoids; and many other components [159]. In comparison to AH, as low as 0.0015% of GH concentration has been reported to enhance the proliferation of corneal keratocytes and CECs while retaining their phenotypical expression [160,161]. 

### 4.5. Dry Eye

Dry eye is a condition with various underlying aetiologies characterized by a disrupted aqueous tear film resulting in symptoms such as mild ocular discomfort or visual disturbance. It is a common condition that affects 5–30% of the population [57]. The use of contact lenses is among the commonest causes of dry eye. About 50% of contact lens wearers reported having dry eye symptoms together with eye discomfort [162]. An inflammatory process has been widely recognized as the fundamental pathophysiology of dry eye disease [163], other than an increase in tear film osmolarity [57]. While tear replacement therapy with artificial tear eye drops has long been the treatment of choice in managing dry eye, this does not aim to reverse the principal mechanism. Therefore, honey with the well-known therapeutic properties of anti-inflammatory and antibacterial has been assessed and evaluated for its potential as a future alternative or adjunctive therapy in managing dry eye disease [164].

Any abnormalities occurring in the lacrimal gland, the gland producing tear film, may also result in tear film disruption. For instance, a long digital device exposure to the gland may cause lacrimal gland hypofunction, and normal aging has resulted in its malfunction, compromising the tear film integrity [164,165]. An evaporative dry eye disease caused by MGD increases tear evaporation and osmolarity, which subsequently extends the susceptibility to ocular surface inflammation and epithelial disruption [166]. 

Optimel Manuka+ Dry Eye drops (Optimel) is an MGH containing a proprietary mix of 16% *Leptospermum* spp. (manuka) honey and other Australian and New Zealand honey, which have been approved to treat chronic dry eye conditions and blepharitis [162]. A study was carried out to investigate the effects of Optimel eye drop on contact lens-related dry eye by comparing two ways of treatment. One group received the Optimel eye drop for two weeks, followed by another two weeks of conventional lubricant (Systane Ultra), while the other group received the reverse pattern of the treatment. Following Optimel eye drop treatment, dry eye symptoms were significantly alleviated. However, the overall signs of dry eyes seemed to have no significant difference, probably due to the limited time of treatment. However, a majority of the subjects reported good compliance with Optimel eye drop treatment, indicating its safety in reducing dry eye symptoms in contact lens wearers [162]. 

In another study, Optimel eye gel (98% of *Leptospermum* species honey) and eye drops were both included as an adjunct to the conventional therapy (warm compression, lid massage, and eye lubricant) in a randomized trial for evaporative dry eye due to MGD. Following two months of therapy, both treatments alleviated conjunctival redness and dry eye symptoms and reduced the need for lubricants. Optimel eye gel significantly improved meibomian gland expressibility and meibum quality. On the other hand, the Optimel eye drop markedly reduced the growth of bacteria at the eyelid margin, whereas both Optimel treatments greatly suppressed *S. epidermidis* growth [166]. In a separate study, Tan J et al. demonstrated the comparison of the effects of an Optimel eye drop and conventional lubricant on the tear film properties in 42 participants with dry eye symptoms (Ocular Surface Disease Index score (ODSI) >12). Following 28 days of treatment, a significant reduction in both tear film evaporation rate and OSDI score were observed in the Optimel eye drop group [163]. 

Manuka honey nasal spray with or without a combination of Optimel eye drops in chronic rhinosinusitis with concurrent dry eye symptoms was reported to cause a significant improvement in nasal symptoms (both groups) and ocular symptoms (eye drop combination group only). Decongestion of the nose and lubrication of the eyes were observed after four weeks of treatment, which were evaluated using the Sino-Nasal Outcome Test and OSDI score, respectively [167]. Presumably, the strong anti-biofilm property of Manuka (*Leptospermum scoparium*) honey [168] might be contributing to the alleviation of chronic rhinosinusitis symptoms as bacterial biofilms are thought to be one of the aetiological factors of chronic rhinosinusitis [169]. As the co-application of manuka eye drop and nasal spray was safe and non-toxic with proven clinical efficacy, a combined use of both has shown a promising potential to relieve chronic rhinosinusitis symptoms with concurrent dry eye symptoms [167]. Figure 3 summarizes the medicinal values of honey in treating ocular disease.

## 5. Medicinal Values of Propolis on Ocular Diseases

Propolis is a natural product that has valuable pharmacological and pharmaceutical properties. With more than 300 biologically active components, propolis has thus far been shown to be effective in treating various ocular diseases in animal and in vitro models [16]. Its efficacy in ocular diseases is likely attributable to its antiglaucoma, antiangiogenic, antioxidant, anti-inflammatory, wound healing agent, and neuroprotective properties [16,17,18,170].

Glaucoma is a disease caused by various factors, including mechanical damage due to increased intraocular pressure or vascular dysregulation that interferes with trabecular meshwork [17]. Previous studies demonstrated that a combination of propolis-acetazolamide can act as a neuroprotector to preserve the normal secondary structure of optic nerve protein. This can be directly connected to the higher content of hydrocarbon chains in propolis [17]. Furthermore, propolis also consists of 18% of triterpenes, which might play a role as an antioxidant agent that reduces intraocular pressure. The increase in intraocular pressure might contribute to oxidative stress, apoptosis, and, finally, glaucoma [171]. In addition, Brazilian green propolis demonstrated neuroprotective effects on retinal ganglion cells through the upregulation of histone acetylation, downregulation of apoptotic stimuli and suppression in NF-ĸB mediated inflammatory pathway in the ischaemic retina in mice [18,172]. In gamma radiation-induced cataracts in Sprague Dawley rats, 80 mg/kg of propolis was found to modulate the antioxidant status (superoxide dismutase, glutathione peroxidase, xanthine oxidase, and malondialdehyde) [173].

The cornea is a transparent, avascular barrier that allows external light to enter the eye. Corneal edema, opacity, and neovascularization are common corneal responses to pathologies, and they have a negative impact on vision quality [174]. The ability of the amniotic membrane to repair corneal defects has long been known. Amniotic membrane is avascular, rich in antiangiogenic factors, inhibits proteinase activity when transplanted to the cornea, and reduces neovascularization and fibrosis while inducing epithelization [175,176]. Previous studies showed that the combination of amniotic membrane and propolis successfully treated subacute alkaline burns of the cornea, accompanied by faster regression of the defect area [170]. 

Additionally, treating lipopolysaccharides-induced uveitis with Turkish propolis significantly reduced ciliary body NF-κB/p65 immunoreactivity and aqueous humor levels of hypoxia-inducible factor-1a (HIF-1α) and TNF-α. Ultrastructural analysis showed fewer vacuoles and reduced mitochondrial degeneration in the retinal pigment epithelium compared to the uveitis group. The intercellular spaces of the inner nuclear layer and outer limiting membrane were comparable with those of the control group; no polymorphonuclear cells or stasis was observed in intravascular or extravascular spaces [177].

CAPE, a phenolic compound isolated from propolis, possesses anti-inflammatory and immunomodulatory properties, demonstrating therapeutic potential in several animal disease models [178]. Furthermore, CAPE-treated mice exhibited a decrease in the ocular infiltration of immune cell populations into the retina; reduced TNF-a, IL-6, and IFN-g serum levels; and inhibited TNF-a mRNA expression in retinal tissues. It was sufficient to suppress cytokine, chemokine, and IRBP-specific antibody production. In addition, the retinal tissues isolated from CAPE-treated EAU mice revealed a decrease in NF-kB p65 and phospho-IkBa [179].

CAPE was also found to be a novel anti-angiogenic agent in vitro, suggesting that it could be used to treat diseases associated with choroidal neovascularization, which may lead to severe visual loss. The hypoxia-induced production of VEGF in the human retinal pigment epithelial cells (ARPE-1) was reduced after pre-treatment with CAPE. This effect was inhibited through the attenuation of reactive oxygen species production and the inhibition of phosphoinositide 3-kinase (PI3K)/AKT as well as hypoxia-inducible factor-1α (HIF-1α) expression [180]. Figure 4 summarizes the medicinal values of propolis in treating ocular disease.

## 6. Methodology

We conducted a literature search to identify recent articles illustrating the medicinal values of honey and propolis in eye diseases. Several online databases were used, including PubMed, Scopus, and ScienceDirect. Any form of publication that was published between 2009 and 2021 was considered. The following keywords were used individually and in combination: honey, propolis, anti-inflammatory, antibacterial, analgesic, anti-nociceptive, eye disease, ocular, ophthalmology, conjunctiva, blepharitis, dry eye, corneal abrasion, corneal ulcer, glaucoma, retina, uvea, and uveitis. Articles found in the reference lists that were relevant were also included. Before being included in this review, all articles were screened.

## 7. Conclusions

In this review, we highlighted the key information regarding the antimicrobial, antioxidant, anti-inflammatory, and analgesic effects of honey and propolis and their associated mechanism of action. Although many studies aimed to discern the mechanism of individual bioactive compounds of honey and propolis, we believed that the synergism of these bioactive compounds contributes to their medicinal benefits. Moreover, we summarized the therapeutic potentials of honey and propolis in various eye diseases. Honey was found to reduce inflammation, inhibit bacterial growth, enhance the healing process, and alleviate dry eye symptoms. On the other hand, propolis has anti-angiogenic properties, lowers intraocular pressure, and inhibits inflammatory responses in addition to providing neuroprotection. It should be acknowledged that the composition of honey and propolis differs depending on the types and commercial brand and does not show similar effects to the eye. Nevertheless, the scarcity of clinical studies suggests that more translational research needs to be performed to attest to the efficacy and safety use of honey and propolis in clinical settings.

## Figures and Tables

**Figure 1 pharmaceuticals-15-01419-f001:**
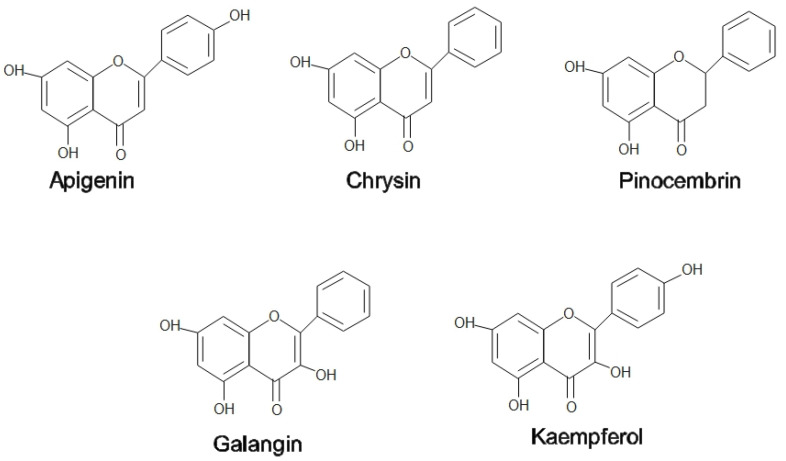
Flavonoids compound in honey and propolis.

**Figure 2 pharmaceuticals-15-01419-f002:**
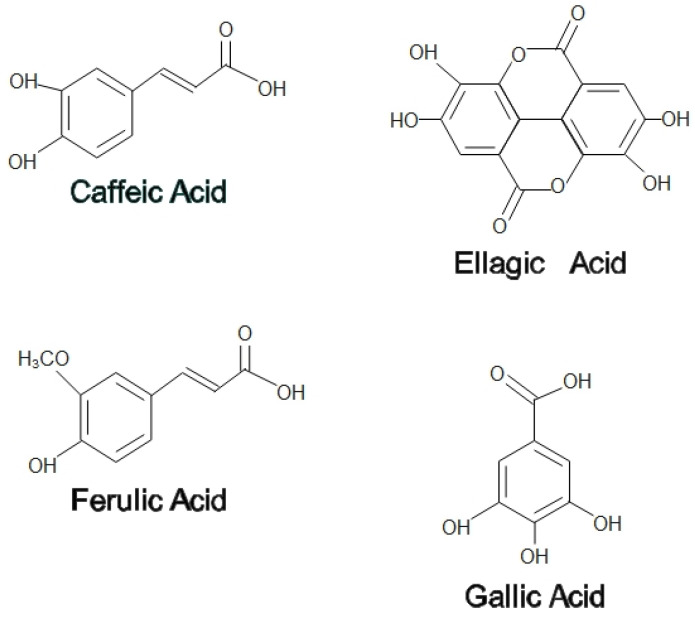
Phenolic acids in honey and propolis.

**Figure 3 pharmaceuticals-15-01419-f003:**
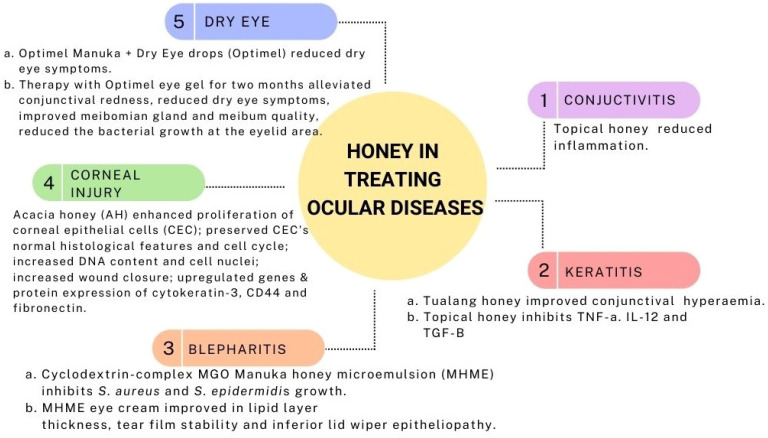
Medicinal values of honey in treating ocular disease.

**Figure 4 pharmaceuticals-15-01419-f004:**
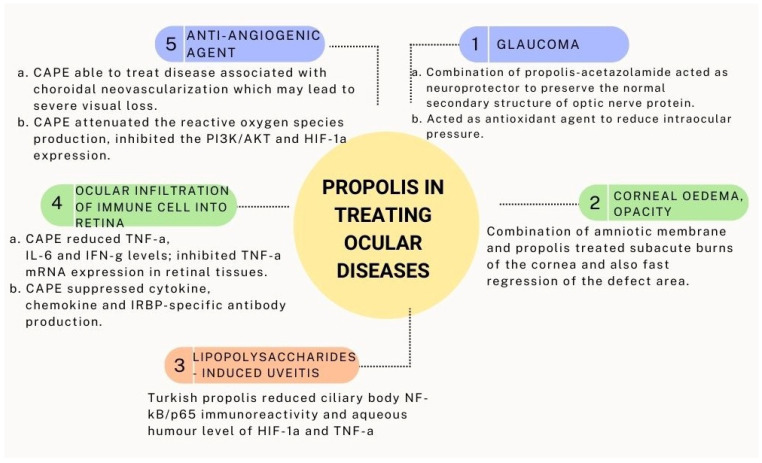
Medicinal values of propolis in treating ocular disease.

**Table 1 pharmaceuticals-15-01419-t001:** Antioxidant signaling pathways of phenolic acids and flavonoids.

Antioxidative Agent	Mode of Study	Outcome	Mechanism of Action (Antioxidant Signaling Pathway)	References
Gallic acid	Sprague-Dawley rats Human type II alveolar epithelial cell line (A549)	↔ IL-6 and TNF-α ↓ lymphocyte and macrophages cell ↓ lipid peroxidation ↑ Increase GSH, SOD, and CAT ↓ ROS production, ↑ in Nrf2, GCL, ERK, and JNK ↓ p38 gene expression activation of Nrf2	Nrf2-antioxidant response element signaling pathway	[24]
chrysin, apigenin, luteolin	rat primary hepatocytes	↑ GSH ↑ GSH: oxidized GSH ratio ↑ HO-1, GCLC, and GCLM gene transcription	Modulating ERK2/Nrf2/ARE signaling pathways	[27]
Gallic acid	Sprague-Dawley rats	↓ lung airspace enlargement ↓ MDA levels ↓ GSH, SOD, and CAT ↑ Nrf2 and HO-1 gene expression ↓ NF-κB gene expression	Modulating Nrf2-HO-1-NF-κB signaling pathways	[25]
Apigenin	Mouse Cardiac fibroblasts	↑ SOD, glutathione peroxidase ↑ miR-122-5p expression ↓ miR-155-5p expression ↓ HIF-1α ↑ c-Ski ↓ TGF-β1-induced Smad2/3 ↑ Smad7	Suppression of NF-κB/TGF-β1	[29]

**Table 2 pharmaceuticals-15-01419-t002:** Physicochemical features of honey and their antibacterial properties.

Physicochemical Feature	Mode of Study	Antimicrobial Properties	Reference
MGO and DHA	Urease activity assay Urease inhibition assay	Urease inhibition which subsequently inhibits ammonia production of bacteria to survive in the acidic environment	[38]
H_2_O_2_ production	Sensitive peroxide/peroxidase assay Broth microdilution assay DNA degradation assays	Oxidative damage causing bacterial growth inhibition and DNA degradation	[39]
High sugar content	Agar-well diffusion Broth macrodilution	Eliminate bacteria through osmotic effects Hinder bacterial growth	[40]
Bee defensins	Modified Lubbock chronic wound biofilm	Antibiofilm activity	[41]

**Table 3 pharmaceuticals-15-01419-t003:** Common flavonoids found in honey and their antibacterial properties.

Flavonoids	Mode of Study	Antimicrobial Properties	Reference
Pinocembrin	In vitro antibacterial activity	Induces cell lysis	[42,43]
Galangin	Minimum inhibitory concentration Growth curve for antimicrobial activity	Bacteriostatic effect via inhibition of murein hydrolase activity	[44]
Quercetin	Antibacterial Evaluation Lipid peroxidation assay	Increase bacterial oxidative cellular stress and limit the availability of L-tryptophan, an essential bacterial growth nutrient	[45]
Apigenin	Antibacterial activity	Modulates nucleic acids processing enzymes (RNA polymerase, DNA gyrase) Alters the bacterial cell wall/membrane synthesis by affecting the synthetic pathway of type II fatty acid and D Alanine ligase	[46,47]
Kaempferol	Antibacterial Mechanism Studies	Destroying bacterial membranes and preventing the development of bacterial resistance	[48]

**Table 4 pharmaceuticals-15-01419-t004:** Common phenolic acids found in honey and their antibacterial properties.

Phenolic Acids	Mode of Study	Antimicrobial Properties	Reference
p-Coumaric acid	ATPase activity Electrophoretic mobility shift assay Spot-test assay	Interfere with the recA protein binding to DNA, subsequently inhibiting bacterial DNA repair mechanism	[49]
Ferulic acid	Agar dilution method Evaluation of changes in intracellular pH, membrane potential, and intracellular ATP concentration	Cellular membrane dysfunction and inhibition of bacterial proliferation	[36]
Gallic acid	Minimum inhibitory concentration Minimum bactericidal concentration Membrane permeabilization Intracellular potassium release Physicochemical surface properties Surface charge	Irreversible disruption in membrane properties (decrease negative surface charge, increase membrane permeability) leading to membrane rupture and intracellular leakage	[50]
Caffeic acid esters	Minimum inhibitory concentrations Minimum bactericidal concentrations Intracellular Reactive Oxygen Species and Glutathione levels	Bactericidal effect through the oxidative stress mechanism	[51]
Ellagic acid	Agar dilution method H. pylori SS1-infected mouse model	Bactericidal properties Inhibiting bacterial colonization	[52]

**Table 5 pharmaceuticals-15-01419-t005:** Analgesic mechanism of various types of honey.

Types of Honey	Mode of Study	Analgesic Mechanism	Reference
**Yemeni Sidr honey**	Acetic Acid-Induced Writhing in Sprague-Dawley rats	Reduced release of inflammatory mediators (NO, PGE2, bradykinin, histamine, serotonin)	[80]
**Mad honey**	Hind paw withdrawal pain in a mice model	Binding of grayonotoxin to the Na channel ++ release of GABA	[81,82]
**Tualang honey**	Tail flick test in Sprague-Dawley rats Clinical studies in post-tonsillectomy patients	Action on opioid receptors Soothing effect	[83,84]
**Nigerian honey**	Hot plate and tail flick tests in mice	Action on opioid receptors	[85]
**Other honey**	Monosodium iodoacetate-induced knee osteoarthritis in female Wistar rats	Reduced release of VEGF	[86]

**Table 6 pharmaceuticals-15-01419-t006:** The antibacterial activity of various propolis with their main constituents.

Propolis	Main constituents	Bacteria	References
**Nepalese propolis (*Apis mellifera* L. and *Trigona* sp.)**	neoflavonoids, isoflavonoids pterocarpans	*Heliobacter pylori* *Staphylococcus aureus* *Shigella flexneri*	[97]
**Chilean propolis**	pinocembrin, apigenin, quercetin, caffeic acid phenethyl ester	*Streptococcus mutans*	[99]
**red, green, and brown propolis**	catechin, ferulic acid, luteolin	*Staphylococcus aureus* *Escherichia coli*	[100]
**green and red propolis**	phenolics, flavonoids	*Staphylococcus aureus*	[101]
**propolis (*Melipona quadrifasciata quadrifasciata* and *Tetragonisca angustula*)**	flavonoids and terpenes	*Staphylococcus aureus* Methicillin-resistant *Staphylococcus aureus Enterococcus faecalis * *Escherichia coli * *Klebsiella. pneumoniae*	[96]
**poplar propolis**	caffeic and p-coumaric acids	*Lactobacillus acidophilus* Oral *streptococci isolates*	[102]
**green propolis**	artepillin-C, kaempferide, drupanin, p-coumaric acid	*Staphylococcus aureus* *Staphylococcus saprophyticus* *Listeria monocytogenes* *Enterococcus faecalis*	[103]
**Serbian propolis**	caffeic acid, quercetin, luteolin, apigenin, p-coumaric acid, kaempferol, naringenin, pinobanksin,	*A. hydrophilia* *Shigella flexneri * *Listeria monocytogenes* *Bacillus subtilis * *Enterococcus faecalis * *Staphylococcus aureus*	[104]
**French poplar propolis**	pinobanksin-3-acetate, pinocembrin, chrysin, galangin, prenyl caffeate	*Staphylococcus aureus* Methicillin-resistant *Staphylococcus aureus* Methicillin-susceptible *Staphylococcus aureus*	[105]
**South African and Brazilian propolis**	chrysin, pinocembrin, galangin, pinobanksin-3-O-acetate.	*Enterococcus faecalis * *Staphylococcus aureus*	[106]
**Brazilian red propolis**	neovestitol, vestitol	*Streptococcus mutans* *Streptococcus sobrinus* *Staphylococcus aureus* *Actinomyces naeslundii*	[107]
**Brazilian propolis**	benzoic acid, diterpenic acids, triterpenic alcohols	*Staphylococcus aureus *	[108]
**Omani propolis**	prenylated flavanones and chalcones	*Staphylococcus aureus * *Escherichia coli*	[109]
**Chilean propolis**	quercetin, myricetin, kaempferol, pinocembrin, coumaric acid, caffeic acid and caffeic acid phenethyl ester	*Streptococcus mutans* *Streptococcus sobrinus*	[99]
**Spanish propolis**	ferulic acid, quercetin	*Staphylococcus epidermidis*	[110]

## Data Availability

Data are contained within the article.

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
