# Peer review of "Therapeutic Potential of Honey and Propolis on Ocular Disease"

_pharmaceuticals, 2022, doi:10.3390/ph15111419_

Round 1
Reviewer 1 Report
1) The title of present manuscript is too long and bit confusing. I suggest the authors to revise it and there is no need to write the sentence after the colon in title.
2) In the abstract section, it is suggested that the authors should describe what they have found in their review article and what are their recommendations.
3) At the end of abstract section there must be a precise conclusion of this study.
4) It would be more important, if the authors describe the key objectives their study at the end of introduction section.
5) It is better if the authors draw the structures in ChemDraw Software and then use in this paper. In present form, these structures are blurred.
6) Table 1 needs major revision as the current data in this table is an ordinary data. It would be better if the authors expand this table by adding more columns and the mode of study, mechanism of action, outcomes, etc.
7) Similar pattern should also be followed in tables 2-4.
8) Authors have described the role of honey in inflammatory mechanisms and their associated diseases. But failed to describe the background of these disorders. I suggest the authors to add at least one paragraph by adding the background of mechanism of induction of inflammatory responses and their associated disorders like diabetes mellitus. For this, I also recommend the authors to consider the following articles (J Biomed Sci. 2016;23(1):87. https://doi.org/10.1186/s12929-016-0303-y; https://doi.org/10.1016/B978-0-323-95120-3.00014-2; Crit Rev Eukaryot Gene Expr. 2017;27(3):229-36. https://doi.org/10.1615/CritRevEukaryotGeneExpr.2017019712).
9) The conclusion of this study in its present form does not reflect the key findings of the study. It should reflect the overall key findings in accordance with future perspectives and its impact on the society.
10) There are several grammatical mistakes and syntax errors. The whole manuscript needs critical revision to remove all the grammatical mistakes and syntax errors.
Author Response
Pharmaceuticals-2006697 (Therapeutic Potential of Honey and Propolis on Ocular Disease)
Response to reviewers
Dear Editor,
Thank you for giving us the opportunity to submit a revised draft of the manuscript “Therapeutic Potential of Honey and Propolis on Ocular Disease” for publication in the Pharmaceuticals Journal. We appreciate the time and effort that you and the reviewer dedicated for providing feedback on our manuscript and are grateful for the insightful comments on and valuable improvement to our paper. We have incorporated most of the suggestions made by the reviewer, and those changes are highlighted in yellow within the manuscript. Please see below, in blue, for a point-by-point response to the reviewers’ comments and concerns. All page numbers refer to the revised manuscript file. The manuscript has been carefully revised to improve the grammar and readability (Highlighted in light blue).
Comments and Suggestions for Authors
REVIEWER #1:
- The title of present manuscript is too long and bit confusing. I suggest the authors to revise it and there is no need to write the sentence after the colon in title.
Author response: We have included this comment accordingly [Page 1, Line 2].
- In the abstract section, it is suggested that the authors should describe what they have found in their review article and what are their recommendations.
- At the end of abstract section there must be a precise conclusion of this study.
Author response: We have revised comments No. 2 and No. 3 accordingly [Page 1, Line 11-28].
- It would be more important, if the authors describe the key objectives of their study at the end of the introduction section.
Author response: We have included this comment accordingly [Page 4, Line 86-89].
- It is better if the authors draw the structures in ChemDraw Software and then use in this paper. In present form, these structures are blurred.
Author response: We have included this comment accordingly [Page 2 and 3, Figure 1 and Figure 2].
- Table 1 needs major revision as the current data in this table is an ordinary data. It would be better if the authors expand this table by adding more columns and the mode of study, mechanism of action, outcomes, etc.
- Similar pattern should also be followed in tables 2-4.
Author response: We have included comments No. 6 and No. 7 accordingly. We found the column for outcomes and mechanism of actions are redundant, so we remain as the; where in Table 2 and 3, we keep the column antimicrobial properties for the mechanism of action, and in Table 5, we keep the column for analgesic mechanism [Page 5 (Table 1); Page 6 (Table 2); Page 7 (Table 3 & Table 4); Page 11 (Table 5)].
- Authors have described the role of honey in inflammatory mechanisms and their associated diseases. But failed to describe the background of these disorders. I suggest the authors to add at least one paragraph by adding the background of mechanism of induction of inflammatory responses and their associated disorders like diabetes mellitus. For this, I also recommend the authors to consider the following articles (J Biomed Sci. 2016;23(1):87. https://doi.org/10.1186/s12929-016-0303-y; https://doi.org/10.1016/B978-0-323-95120-3.00014-2; Crit Rev Eukaryot Gene Expr.2017;27(3):229-36. https://doi.org/10.1615/CritRevEukaryotGeneExpr.2017019712).
Author response: We have included this comment accordingly [Page 9, Line 202-205; Line 208-217].
- The conclusion of this study in its present form does not reflect the key findings of the study. It should reflect the overall key findings in accordance with future perspectives and its impact on the society.
Author response: We have included this comment accordingly [Page 24, Line 732-745]
- There are several grammatical mistakes and syntax errors. The whole manuscript needs critical revision to remove all the grammatical mistakes and syntax errors.
Author response: We have corrected the grammatical mistakes and syntax errors which are highlighted in light blue.
REVIEWER #2:
- Authors are advised to go through the whole manuscript for spell check and grammar.
Author response: We have corrected the grammatical mistakes and spell check which are highlighted in light blue.
- The references should also be checked out and need to add doi of references.
Author response: We have included this comment accordingly [These paper does not provide doi number. Therefore, we provide PMID number].
Page 11, Line 800
Page 18, Line 1145; Line 1150
Page 19, Line 1159; Line 1160; Line 1173; Line 1189
Doi for references
Page 12, Line 869, Line 872
Page 15, Line 1018; Line 1023; Line 1029
Page 17, Line 1088
Page 20, Line 1212
Please see the attachment.

Reviewer 2 Report
The work is well-designed and appropriate for publication. The authors have justified the title. Authors are advised to go through the whole manuscript for spell check and grammar. The references should also be checked out and need to add doi of references.
Round 2
Reviewer 1 Report
All the comments have been addressed in an appropriate manner. I have no further comments.